# Point-of-care ultrasound reveals extensive pathology in Gabonese preschool-age children with urogenital schistosomiasis

Jonathan Remppis[1,2,3,4]*, Anais Verheyden[3], Ayten Sultanli[1,2], Amaya Lopez Bustinduy[5], Tom Heller[6], Ayola Akim Adegnika[1,3], Martin Peter Grobusch[1,2,3,7], Michael Ramharter[3,8,9], Elizabeth Joekes[10], Sabine Bélard[1,2,3]

1 Institute of Tropical Medicine, University of Tübingen, Tübingen, Germany, 2 German Center for Infection Research (DZIF), Partner Site Tübingen, Tübingen, Germany, 3 Centre de Recherches Médicales de Lambaréné (CERMEL), Lambaréné, Gabon, 4 General Pediatrics, Pediatric Neurology and Developmental Medicine, University Children's Hospital Tübingen, Tübingen, Germany, 5 Department of Clinical Research, London School of Hygiene and Tropical Medicine, London, United Kingdom, 6 Lighthouse Clinic, Kamuzu Central Hospital, Lilongwe, Malawi, 7 Center of Tropical Medicine and Travel Medicine, Department of Infectious Diseases, Amsterdam University Medical Centers, location AMC, Amsterdam, The Netherlands, 8 Department of Tropical Medicine, Bernhard Nocht Institute for Tropical Medicine and I. Department of Medicine, University Medical Center Hamburg-Eppendorf, Hamburg, Germany, 9 German Center for Infection Research, Partner Site Hamburg-Lübeck-Borstel-Riems, Hamburg, Germany, 10 Department of Clinical Sciences, Liverpool School of Tropical Medicine, Liverpool, United Kingdom

* jonathan.remppis@med.uni-tuebingen.de

**Editor:** jong-Yil Chai, Seoul National University College of Medicine, KOREA, REPUBLIC OF

## Abstract

### Background

Historically, urinary tract pathology caused by *S. haematobium* infection was thought to affect predominantly school-age children (SAC) and adults. Increasing availability of ultrasound data from endemic areas demonstrates that even younger children develop potentially irreversible pathology.

### Methodology/principal findings

Point-of-care ultrasound for urinary schistosomiasis and urine microscopy were performed across age groups in 105 patients with symptomatic urogenital schistosomiasis in Lambaréné, Gabon. Of 96 ultrasound scans with sufficient image quality, bladder wall thickening > 5mm was found in 9/20 (45%) preschool-age children (PSAC), 29/51 (57%) SAC and 7/25 (28%) adults. Upper urinary tract pathology was found in 19/90 (21%) patients across age groups, up from three years of age. Urine egg counts were highest in PSAC, with high-intensity infection (≥ 50 eggs/10 ml urine) in 19/24 (79%) and hyper-infection (≥ 500 eggs/10 ml urine) in 10/24 (42%). Bladder wall thickening > 5mm and upper urinary tract pathology correlated significantly with high-intensity infection with crude odds ratios of 8.6 (95% CI 3.1-23.8; p<0.001) and 6.6 (95% CI 1.4-30.7; p=0.02), respectively. Three months after praziquantel treatment, parasitology showed a cure rate of 51% and egg reduction rate of 95%, while bladder wall thickening and upper urinary tract pathology persisted in 12/41 (29%) and 7/12 (58%) patients.

**Data availability statement:** All relevant data are within the manuscript and its Supporting Information files.

**Funding:** JR, AS and SB were partially funded by the German Center for Infection Research (DZIF) during data analysis and manuscript writing (grant number: CTP TTU 03.702). https://www.dzif.de. The funders had no role in study design, data collection and analysis, decision to publish, or preparation of the manuscript.

**Competing interests:** The authors have declared that no competing interests exist.

## Conclusions/significance

A high proportion of PSAC in areas endemic for urogenital schistosomiasis already have detectable urinary tract pathology. Our findings highlight the urgent need to include this age group in mass drug administration programs, as recommended now by WHO. Further, particular attention should be paid to individual patient care.

### Author Summary

Urogenital schistosomiasis is a neglected tropical disease that can substantially harm the urinary tract. Growing availability of ultrasound data in regions where urogenital schistosomiasis is prevalent indicates that the disease causes relevant urinary tract pathology already in very young children. In our study, we used ultrasound and urine microscopy to investigate 105 patients across age groups with urogenital schistosomiasis in Lambaréné, Gabon. We found thickening of the urinary bladder wall in a high proportion of preschool-age children. Damage of the upper urinary tract (kidneys or ureters) was detected in children as young as three years old. Infection intensity, measured by urine egg count, was highest in preschool-age children, and patients with higher egg counts were more likely to show pathology of the urinary tract. In some patients, pathology persisted three months after antiparasitic treatment. Our findings show that very young children can be affected by urogenital schistosomiasis and therefore require special attention in disease control programs and individual patient care.

## Introduction

Urogenital schistosomiasis (UGS) is a waterborne parasitic disease mainly resulting from infection with *Schistosoma haematobium*. It belongs to the neglected tropical diseases and affects almost 240 million people worldwide [1]. Untreated infection causes substantial morbidity of both the urinary and genital tract in endemic areas. Urinary tract pathology occurs in the early stage of infection, mainly affects the bladder wall and distal ureters and is often reversible after treatment [2]. In chronically infected patients, UGS can cause persistent bladder wall changes, irreversible hydronephrosis with a risk for kidney failure as well as squamous cell carcinoma of the bladder [3–5]. Genital schistosomiasis (GS) is a still underrated manifestation of *S. haematobium* infection, which can lead to sexual dysfunction and infertility in both females and males and increases the risk for ectopic pregnancies [6].

In Gabon, UGS is endemic and preventive chemotherapy is generally recommended by the WHO [7]. The local epidemiology differs substantially between provinces and different areas, with *S. haematobium* prevalence estimates ranging from 0.8 to 45% in school-age children [8]. While data on the epidemiology of UGS has increased during the last two decades, most of it is limited to parasitological determinants. Evaluation of morbidity has been rarely undertaken, as access to diagnostic tools such as ultrasound was limited; thus, little remains known about the prevalence and extent of associated urinary tract pathology. However, morbidity data across the age ranges is needed to estimate the disease burden for the local population and to guide local control strategies.

Our study aimed to detect urinary tract pathology in symptomatic UGS patients in an endemic area and to correlate the findings with demographic, clinical, and parasitological factors.

## Methods

This prospective study was conducted at the Centre des Recherches Médicales (CERMEL) located in Lambaréné, a 40.000 inhabitants' town in Gabon, Central Africa. Multiple studies on various aspects of UGS have been previously conducted at the CERMEL, and local hotspots with high prevalence of *S. haematobium* infections have been identified in Lambaréné and its vicinity [8].

### Ethical approval

This study was approved by the scientific review committee and the institutional ethics committee of the Centre de Recherches Médicales de Lambaréné (approval number 015/2015). Written informed consent was obtained from the participants or their legal guardians.

### Study preparations

The research project was designed as a pilot study focusing on the development and evaluation of a point-of-care ultrasound (POCUS) protocol called *Focused Assessment with Sonography for Urinary Schistosomiasis (FASUS),* for the detection of UGS-related pathology. Details on protocol development, ultrasound training, and evaluation of the protocol have been published separately [9]. In brief, the novel POCUS protocol comprises longitudinal and transverse scans of the urinary bladder and both kidneys to evaluate for bladder wall abnormalities, distal and proximal ureter dilatation, and hydronephrosis.

### Recruitment procedures

Inclusion criteria of the study were previous or ongoing macro-hematuria; exclusion criterion was known disease of the urinary tract related to causes other than UGS. Study participants of all ages were recruited by convenience sampling in those areas of Lambaréné that had been previously identified as hotspots for UGS. These include the quarters 'Petit Paris 3', 'Route Fangui', and 'Moussamoukougou'. This recruitment strategy was chosen to maximize the number of participants fulfilling the inclusion criteria; and therefore limit the study cohort to participants with clinical symptoms of UGS.

### Clinical and parasitology procedures

After completion of a clinical questionnaire, three urine samples were collected for urine dipstick testing as well as urine filtration (10ml urine) for microscopic egg detection of *S. haematobium*. A positive result was defined as any detection of eggs in the filtered urine. High-intensity infection was defined as ≥ 50 eggs/10ml urine according to the WHO [10]. Because egg counts were more than ten-fold higher in parts of our study population and technical reasons led to an upper count limit of 500/10ml, we introduced the additional category 'hyper-infection' for egg counts of ≥ 500/10ml, as a sub-category of high-intensity infection.

### Focused assessment with sonography for urinary schistosomiasis

POCUS evaluation was done for all participants, either at the participants' residence or at CERMEL. Two locally trained ultrasound operators performed the scans, using a portable ultrasound device (MINDRAY Digital Ultrasonic Diagnostic Imaging System model DP-10, Mindray Medical International Limited, Shenzhen, China) with a curved array transducer (MINDRAY model 35C50EB). Digital video clips and still images were uploaded on a teleradiology platform for remote expert review, as described in detail before [9]. The results reported below are based on the interpretation by the ultrasound experts, not the operators.

### Treatment and follow-up

Study participants diagnosed with UGS by microscopy were treated orally with a single dose of 40mg/kg praziquantel. Follow-up visits at the participants' residence were attempted one month (M1, 28 ± 2 days) and three months (M3, 90 ± 10 days) after treatment and comprised assessment of clinical symptoms and urine analysis by dipstick and microscopy on two consecutive days. Follow-up POCUS with remote expert review was performed on participants in whom the initial POCUS examination had detected urinary tract pathology. Participants with persistent egg excretion or sonographic findings of UGS three months after treatment were re-treated with a single dose of 40 mg/kg praziquantel. In cases where advanced UGS-related pathology was not responsive to treatment, the participant was referred for urological care.

### Data analysis

Data were entered into OpenClinica version 3.0.4 (OpenClinica, Boston, MA, USA). Data analysis was performed using Microsoft Excel (Microsoft, Redmond, WA, USA). Statistical analysis was performed with IBM SPSS Statistics 28 (IBM, Armonk, NY, USA) and comprised crude risk factor analysis with chi-square test and logistic regression. Age groups were defined as follows; preschool-age children (PSAC): < 6 years; school-age children (SAC): 6-15 years; adults: ≥ 16 years.

## Results

### Study flow

Between December 2015 and June 2016, 119 participants were recruited. One participant withdrew consent after enrolment. Of the remaining participants, 105/118 (89%) were microscopically diagnosed with UGS by *S. haematobium* infection. Of these, 95/105 (90%) underwent praziquantel treatment under study conditions. M1 follow-up was performed in 62/95 (65%) participants, and M3 follow-up in 70/95 (74%) participants; thus, some patients that had been missed at M1 follow-up could be relocated for M3 follow-up (Fig 1).

### Participants

Data on demographic, clinical, parasitological and ultrasound findings at baseline are presented in Table 1. Median age was 11 (IQR 6.5; 16.2, range 2 - 46) years. Median reported duration of exposure to freshwater was 4.0 (IQR 1.5, 6.0) years. In participants reporting previous praziquantel treatment, median duration since the last treatment was 1.0 (IQR 0.8, 3.0) year. In those reporting ongoing hematuria, median duration of hematuria was 4.0 (IQR 1.3, 13.0) months.

### Urine analysis

In 101/105 (96%) of participants with confirmed UGS, *S. haematobium* eggs were detected in the first collected urine sample; in four patients, three samples were needed to detect the infection. High-intensity infection (≥ 50 eggs/10ml urine) was present in 69 (66%) participants; 37 (35%) had hyper-infection (≥ 500/10ml). Crude risk factors for high-intensity infection were PSAC (OR 3.8; 95% CI 1.1-13.0; p=0.03) and reported ongoing hematuria (OR 5.9; 95% CI 2.0-17.5; p=0.001). Crude risk factors for hyper-infection were PSAC (OR 9.3; 95% CI 1.8-48.4; p=0.008), SAC (OR 11.6; 95% CI 2.5-53.9; P=0.002), and reported ongoing hematuria (OR 13.0; 95% CI 1.7-101.5; p=0.02) (Table 2).

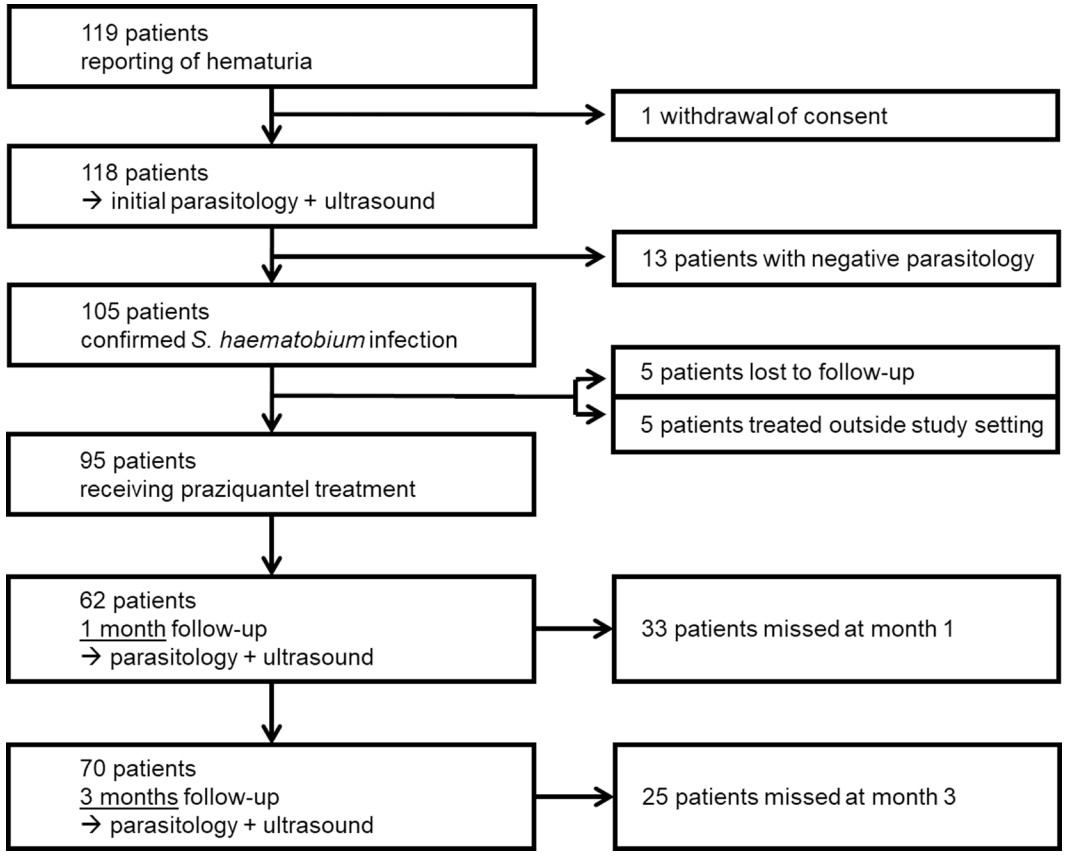

**Fig 1. Study flow.**

## Sonographic findings

Baseline point-of-care ultrasound was of sufficient image quality to allow interpretation of the bladder by the experts in 96/105 (91%) and interpretation of the kidneys in 103/105 (98%) participants. The complete upper urinary tract (both kidneys, proximal and distal ureters) was sufficiently displayed in 90/105 (86%) participants. Sonographic findings by age are presented in Figs 2 and 3. An example of advanced pathology in a 3-year-old boy is presented in Fig 4.

Crude risk factors for presence of bladder wall thickening >5mm were SAC (OR 3.4; 95% CI 1.2-9.5; p=0.02) and high-intensity infection (OR 8.6; 95% CI 3.1-23.8; p<0.001). Crude risk factor for upper urinary tract pathology was high-intensity infection (OR 6.6; 95% CI 1.4-30.7; p=0.02) (Table 2).

## Resolution after treatment

Ninety-five patients received praziquantel treatment by the study team and met the requirements for follow-up analysis. Age and sex distribution were comparable to the entire study population. Among the 95 patients, 42 had bladder wall thickening ≥ 5mm, and 17 had upper urinary tract pathology before treatment. Resolution of egg excretion and urinary tract pathology after treatment is presented in Fig 5.

Three months after treatment, urinary egg excretion was still detected in 34 patients, which equates to a the cure rate of 36/70 (51%); the mean egg count had decreased from 255 to 12 eggs/10ml urine, corresponding to a 95% egg reduction rate. Crude risk factors for persistence

Table 1. Baseline data of participants with confirmed *S. haematobium* infection.

| | Total | PSAC (< 6 years) | SAC (6-15 years) | Adults (≥ 16 years) |
|---|---|---|---|---|
| **Total (%)** | **105 (100)** | **24 (100)** | **53 (100)** | **28 (100)** |
| **Female, n (%)** | 50 (48) | 8 (33) | 25 (47) | 17 (61) |
| **Previous PZQ treatment, n (%)** | 43 (41) | 3 (13) | 26 (49) | 14 (50) |
| **Reported symptoms, n (%)** | | | | |
| **Ongoing hematuria** | 86 (82) | 21 (88) | 46 (87) | 19 (68) |
| **Previous hematuria** | 19 (18) | 3 (13) | 7 (13) | 9 (32) |
| Dysuria* | 60 (57) | 14 (58) | 29 (55) | 17 (61) |
| **Recurrent UTIs** | 2 (2) | 0 | 0 | 2 (7) |
| **Symptoms in males** | | | | |
| **Testicular swelling** | 6/55 (11) | 1/16 (6) | 4/28 (14) | 1/11 (9) |
| **Symptoms in females** | | | | |
| **Genital itching** | 24/50 (48) | 5/8 (63) | 8/25 (32) | 11/17 (65) |
| **Urinary incontinence** | 13/50 (26) | 0 | 8/25 (32) | 5/17 (29) |
| **Malodorous vaginal discharge** | 8/50 (16) | 0 | 2/25 (8) | 6/17 (35) |
| **Urine analysis, n (%)** | | | | |
| **Hematuria on dipstick** | 98 (93) | 22 (92) | 50 (94) | 26 (93) |
| **1 - 49 eggs/10ml** | 36 (34) | 5 (21) | 17 (32) | 14 (50) |
| **≥ 50 eggs/10ml (high-intensity infection)** | 69 (66) | 19 (79) | 36 (68) | 14 (50) |
| **≥ 500 eggs/10ml (hyper-infection)** | 37 (35) | 10 (42) | 25 (47) | 2 (7) |
| **Sonographic findings, n# (%)** | | | | |
| **Bladder wall thickening > 5mm** | 45/96 (47) | 9/20 (45) | 29/51 (57) | 7/25 (28) |
| **Upper urinary tract pathology** | 19/90 (21) | 4/20 (20) | 13/46 (28) | 2/24 (8) |
| **Distal ureter dilatation** | 13/91 (14) | 4/20 (20) | 9/47 (19) | 0/24 |
| **Unilateral hydronephrosis** | 5/103 (5) | 1/24 (4) | 3/52 (6) | 1/27 (4) |
| **Bilateral hydronephrosis** | 2/103 (2) | 0/24 | 2/52 (4) | 0/27 |

*PSAC, preschool-age children; SAC, school-age children; PZQ, praziquantel; UTI, urinary tract infection; *ongoing or previous; #denominator is number of scans with sufficient image quality for interpretation*

of egg excretion were SAC (OR 6.2; 95% CI 1.7-22.8; p=0.006) and reported ongoing hematuria at baseline (OR 7.0, 95% CI 1.4-34.7; p=0.02) (Table 2). Risk factors for persistence of pathology were not investigated due to small case numbers.

## Discussion

Our study provides a detailed report on urinary tract pathology related to UGS in Gabon for the first time. By including patients across different age groups up from two years of age, our data yields important insights into the age distribution of morbidity caused by UGS. Our results should primarily guide local schistosomiasis control; but can also add valuable information to control strategies and individual patient care on a global level.

### High morbidity in PSAC

Most PSAC in our study cohort were heavily infected, as per egg load; and more than half exhibited bladder wall pathology (i.e. irregularities or thickening), with a rapid increase of pathology by the age of four years. Of note, by the years 2015 - 2016, when our study was conducted, the WHO did not generally recommend preventive chemotherapy for PSAC: WHO recommendations from 2001 targeted on SAC in endemic areas, and in a progress

**Table 2. Risk factor analysis (crude).**

| Investigated risk factors | Urine Egg count ≥ 50/ 10ml | | Urine egg count ≥ 500/ 10ml | | Bladder wall thickening > 5mm | | Any upper urinary tract pathology | | Egg excretion persistent 3 months after treatment | |
|---|---|---|---|---|---|---|---|---|---|---|
| | OR (95% CI) | p | OR (95% CI) | p | OR (95% CI) | p | OR (95% CI) | p | OR (95% CI) | p |
| **Age group** | | | | | | | | | | |
| PSAC vs adults | 3.8 (1.1-13.0) | **0.03** | 9.3 (1.8-48.4) | **0.008** | 2.1 (0.6-7.3) | 0.24 | 2.8 (0.5 -15.9) | 0.275 | 2.7 (0.6-12.0) | 0.187 |
| SAC vs adults | 2.1 (0.8-5.4) | 0.12 | 11.6 (2.5-53.9) | **0.002** | 3.4 (1.2-9.5) | **0.02** | 4.3 (0.9-21.1) | 0.07 | 6.2 (1.7-22.8) | **0.006** |
| **Sex** | | | | | | | | | | |
| female vs male | 1.2 (0.5-2.7) | 0.64 | 0.9 (0.4-2.0) | 0.80 | 1.06 (0.48-2.4) | 0.88 | 1.0 (0.4-2.7) | 0.97 | 2.2 (0.9-5.8) | 0.098 |
| **Reported macrohematuria** | | | | | | | | | | |
| ongoing vs previous | 5.9 (2.0-17.5) | **0.001** | 13.0 (1.7-101.5) | **0.02** | 2.0 (0.7-5.9) | 0.21 | 4.4 (0.5-36.0) | 0.17 | 7.0 (1.4-34.7) | **0.02** |
| duration ≥ 1 vs < 1 year | 2.2 (0,7-6.8) | 0.16 | 1.50 (0.6-3.7) | 0.39 | 1.3 (0.5-3.2) | 0.63 | 2.6 (0.9-7.6) | 0.09 | 2.4 (0.7-8.1) | 0.16 |
| **Previous treatment** | | | | | | | | | | |
| never vs ≤ 1 year ago | 2.0 (0.8-5.6) | 0.19 | 1.9 (0.7-5.6) | 0.23 | 2.1 (0.7-5.9) | 0.18 | 2.7 (0.5-13.2) | 0.22 | 1.4 (0.3-6.09 | 0.63 |
| > 1 vs ≤ 1 year ago | 1.0 (0.3-3.4) | 1.0 | 0.6 (0.1-2.5) | 0.47 | 1.5 (0.4-5.5) | 0.55 | 2.8 (0.4-18.0) | 0.27 | 0.8 (0.2-4.4) | 0.83 |
| **Freshwater contact** | | | | | | | | | | |
| > 2 years vs. ≤ 2 years | 1.8 (0.7-5.2) | 0.23 | 1.60 (0.5-4.6) | 0.40 | 2.1 (0.8-6.0) | 0.15 | 1.5 (0.36.2) | 0.61 | NA | NA |
| **Eggs/10ml at baseline** | | | | | | | | | | |
| ≥ 50 vs < 50 | NA | NA | NA | NA | 8.6 (3.1-23.8) | **<0.001** | 6.6 (1.4-30.7) | **0.02** | 2.5 (0.9-6.8) | 0.08 |
| ≥ 500 vs < 500 | NA | NA | NA | NA | 10.3 (3.6-29.0) | **<0.001** | 4.7 (1.6-13.7) | **0.01** | 3.0 (0.9-10.2) | 0.08 |

*PSAC, preschool-age children; SAC, school-age; OR, Odd's ratio; CI, Confidence interval; NA, not applicable.*

report released 2012, a possible age extension to PSAC was discussed but not generally recommended [11,12]. It was only in 2022 that the WHO recommended preventive chemotherapy for children from two years in endemic communities, due to increasing evidence that chronic pathology already affects children before school-age [13–15]. Our findings underpin that this recently introduced age expansion is necessary to prevent potentially irreversible morbidity in the youngest. This applies not only for urogenital but also for intestinal schistosomiasis, where incipient fibrotic changes that were mostly attributed to older populations have been recently detected in PSAC [16].

## High infection intensity

Our data point at a high *S. haematobium* infection intensity in hotspots for schistosomiasis around Lambaréné; particularly in PSAC and SAC. The rate of high-intensity infection in our study (69/105, 66%) was higher compared to a concomitantly performed study on children and young adults in the same area (46/103, 45%), presumably due to different recruitment strategies [17]. Our findings underline the necessity of effective local control programs. While the WHO already had recommended regular MDA of Praziquantel for SAC in the study area prior conduction of the study, our data shows that only half of these children had actually

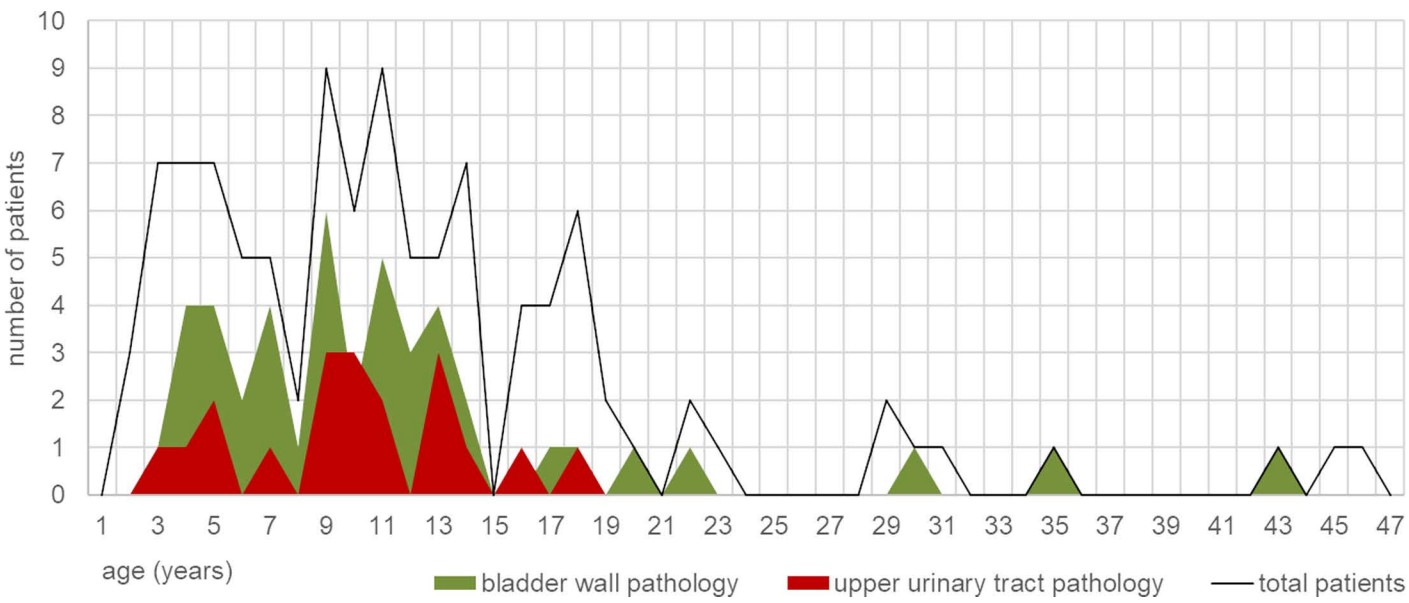

**Fig 2. Sonographic findings by age.**

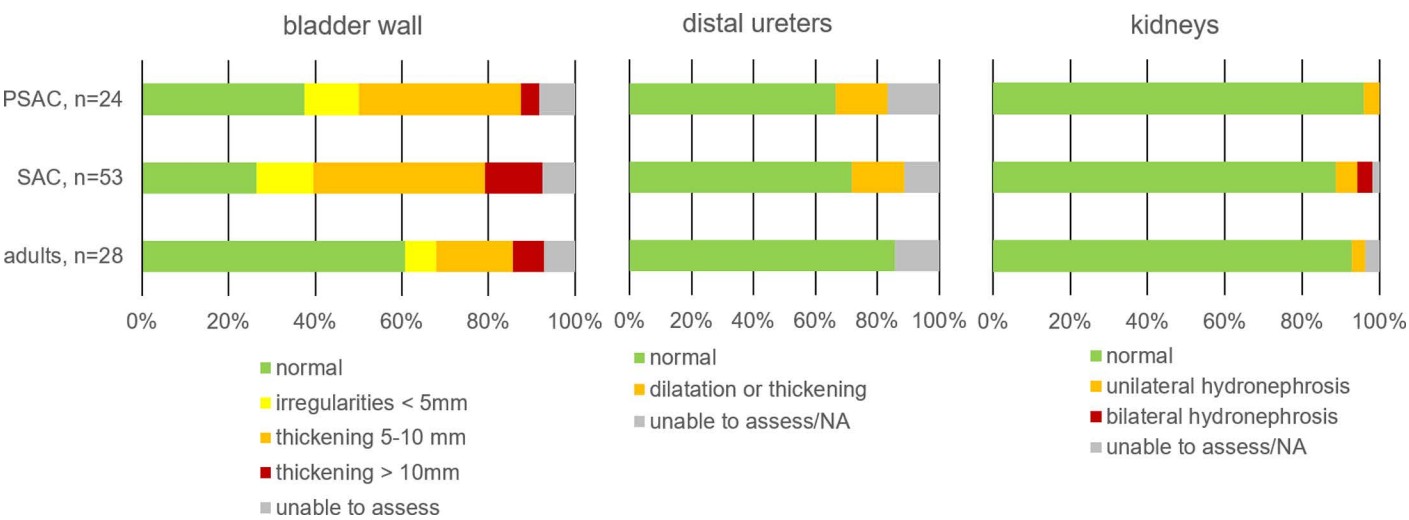

**Fig 3. Sonographic findings by age group. PSAC, preschool-age children; SAC, school-age children; NA, not available.**

been previously treated – thus, the implementation of local control programs seems to be insufficient; in the 2022 WHO guideline, MDA in Gabon is still considered 'not at scale or irregular' [15].

## Correlation between infection intensity and pathology

As in other studies, the presence of urinary tract pathology correlated highly positively with infection intensity [18,19]. In settings with available parasitology but restricted access to ultrasound, patients with heavy infection should therefore be prioritized by means of referral to other health care centers with the possibility of ultrasound examination. This applies

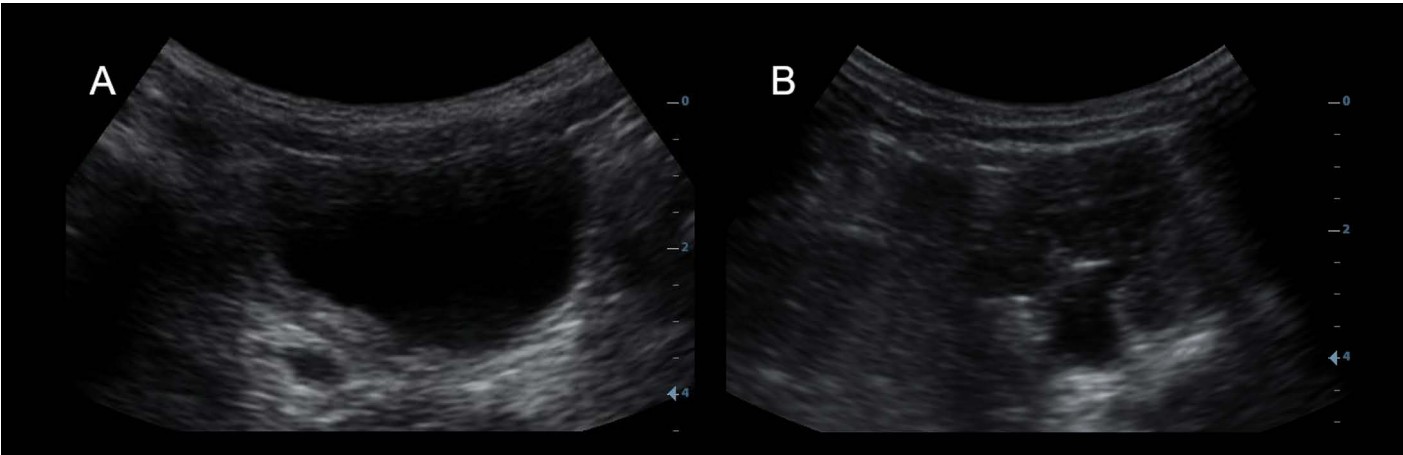

**Fig 4. Urinary tract pathology in a 3-year-old boy detected by point-of-care ultrasound. (A) Bladder wall thickening and distal ureter dilatation. (B) Dilatation of left kidney pelvis (recorded after bladder voiding).**

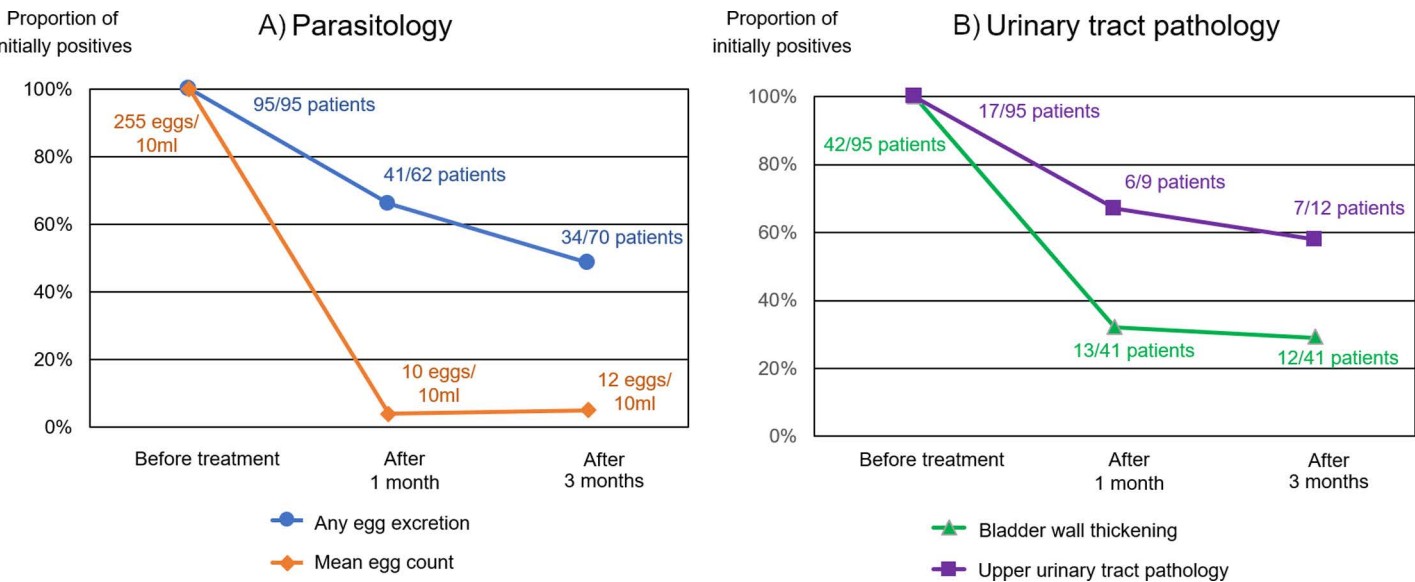

**Fig 5. Resolution after Praziquantel treatment. (A) Resolution of S. haematobium egg excretion after a single dose of 40mg/kg Praziquantel. (B) Resolution of urinary tract pathology after a single dose of 40mg/kg Praziquantel.**

particularly for SAC with ongoing hematuria, which is the group with the highest risk of potentially irreversible urinary tract pathology. If advanced pathology is detected by ultrasound, therapeutic options comprise repeated praziquantel treatment and, if available, surgical procedures such as laser endo-ureterotomy, stenting or ureteral re-implantation [20].

### Treatment outcome

The egg reduction rate after a single dose of PZQ in our study was comparable to a concomitantly performed longitudinal study in Gabon (95% vs. 93%) while the cure rate was lower (36/70, 51% vs. 52/67, 78%); probable reason is the high proportion of heavily infected

participants in our study, which has been identified as risk factor for treatment failure in the same study [17]. Another reason might be prompt re-infection, as most study participants had continuous freshwater contact after treatment.

Our data shows that the majority of bladder wall pathology was reversible after treatment, while upper urinary tract pathology persisted after three months in most study participants. This has been similarly described in other studies [18,19]. The overall resolution of pathology three months after treatment was low in our study; possible reasons include the short follow-up period, as resolution of pathology after a single dose of PZQ might take up to six months [19]. Another reason might be the increased sensitivity in detection of bladder wall pathology in our study, with remote review by two ultrasound experts; however, over-interpretation of pathology is also possible.

### Genital schistosomiasis

Almost half of our female participants reported symptoms compatible with GS. Although these symptoms are non-specific, our findings might point at the presence of GS, which is a neglected disease in Gabon, only described by few reports [21,22]. As GS harbors the risk of serious complications such as infertility, ectopic pregnancy and hemoperitoneum, further studies on its local epidemiology should be performed to inform control programs and individual patient care [6].

### Parasitology diagnostics

In scientific literature, an egg count of $\geq 50/10$ml is the established definition of high-intensity infection [10]. Of note, a high proportion of our study population had egg counts ten-fold exceeding this cutoff value, and risk factors differed when stratifying for higher egg counts. Accordingly, the definition of an additional category (such as 'hyper-infection') might be a useful approach for scientific analyses or clinical algorithms in highly endemic areas.

In most of our participants, one urine sample sufficed for the diagnosis of *S. haematobium* infection by microscopy. The collection of additional urine samples led to egg detection in only four participants. Thus, microscopic examination of a single urine sample seems to have a sufficient diagnostic yield in populations with high parasite loads, which is in line with the current WHO manual for impact assessments [23].

### Limitations

As our study was primarily designed as a pilot study for the evaluation of a new ultrasound method, our study population was highly selected. Parasite counts and pathology rates are therefore not representative of the entire local population; but depict the specific situation of people living in highly endemic areas and reporting of hematuria. Real egg reduction rates are probably higher than reported, because egg counts of more than 500/10ml at baseline were rounded down, resulting in a lower arithmetic mean at baseline. The reference test of a remote expert review of ultrasound clips for detection of urinary tract pathology is inferior to a locally performed expert reference scan; but was the only feasible option in this particular study setting.

### Conclusions

UGS is the predominant cause for macrohematuria in the study area. High egg loads and substantial urinary tract pathology point at chronic infection and insufficient regular praziquantel chemotherapy as recommended by WHO [15]. The extent of relevant and potentially irreversible pathology in PSAC should be a warning that this age group must no longer be neglected.

Annual preventive treatment should be administered to anyone aged two years and upwards in endemic areas of Gabon and worldwide.

## Supporting information

**S1 Data.** Database comprising clinical, radiological and laboratory data collected at the different study time points (inclusion, M1 follow-up, M3 follow-up). M, male; F, female; NA, not available; UTI, urinary tract infection; UGS, urogenital schistosomiasis; FASUS, Focused Assessment with Sonography for Urinary Schistosomiasis; TS, transversal scan; LS, longitudinal scan; Prox, proximal; M1, month 1; M3, month 3; FU, follow up.
(XLSX)

## Author contributions

**Conceptualization:** Jonathan Remppis, Amaya Lopez Bustinduy, Tom Heller, Elizabeth Joekes, Sabine Bélard.

**Formal analysis:** Jonathan Remppis, Ayten Sultanli.

**Funding acquisition:** Sabine Bélard.

**Investigation:** Jonathan Remppis, Anais Verheyden, Elizabeth Joekes, Sabine Bélard.

**Methodology:** Jonathan Remppis, Amaya Lopez Bustinduy, Tom Heller, Ayola Akim Adegnika, Martin Peter Grobusch, Michael Ramharter, Elizabeth Joekes, Sabine Bélard.

**Project administration:** Jonathan Remppis, Sabine Bélard.

**Supervision:** Amaya Lopez Bustinduy, Ayola Akim Adegnika, Martin Peter Grobusch, Michael Ramharter, Sabine Bélard.

**Visualization:** Jonathan Remppis.

**Writing – original draft:** Jonathan Remppis, Sabine Bélard.

**Writing – review & editing:** Jonathan Remppis, Anais Verheyden, Ayten Sultanli, Amaya Lopez Bustinduy, Tom Heller, Ayola Akim Adegnika, Martin Peter Grobusch, Michael Ramharter, Elizabeth Joekes, Sabine Bélard.

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
