## [Decision Letter · Decision Letter 0]

3 Feb 2025

PNTD-D-24-01698Point-of-care ultrasound reveals extensive pathology in Gabonese preschool-age children with urogenital schistosomiasisPLOS Neglected Tropical Diseases Dear Dr. Jonathan Remppis, Thank you for submitting your manuscript to PLOS Neglected Tropical Diseases. After careful consideration, we feel that it has merit but does not fully meet PLOS Neglected Tropical Diseases's publication criteria as it currently stands. Therefore, we invite you to submit a revised version of the manuscript that addresses the points raised during the review process especially the discussion about underage children concerning by a reviewer. Please submit your revised manuscript within 30 days. If you will need more time than this to complete your revisions, please reply to this message or contact the journal office at plosntds@plos.org. Please include the following items when submitting your revised manuscript: * A rebuttal letter that responds to each point raised by the editor and reviewer(s). You should upload this letter as a separate file labeled 'Response to Reviewers '. This file does not need to include responses to any formatting updates and technical items listed in the 'Journal Requirements' section below.* A marked-up copy of your manuscript that highlights changes made to the original version. You should upload this as a separate file labeled 'Revised Manuscript with Track Changes '.* An unmarked version of your revised paper without tracked changes. You should upload this as a separate file labeled 'Manuscript '. If you would like to make changes to your financial disclosure, competing interests statement, or data availability statement, please make these updates within the submission form at the time of resubmission. Guidelines for resubmitting your figure files are available below the reviewer comments at the end of this letter. We look forward to receiving your revised manuscript. Kind regards, Wannaporn Ittiprasert, Ph.DAcademic EditorPLOS Neglected Tropical Diseases Jong-Yil ChaiSection EditorPLOS Neglected Tropical Diseases

Shaden Kamhawi

co-Editor-in-Chief

Paul Brindley

co-Editor-in-Chief

**Journal Requirements:**

At this stage, the following Authors/Authors require contributions: J. Remppis, A. Verheyden, A. Sultanli, A.L. Bustinduy, T. Heller, A.A. Adegnika, M.P. Grobusch, M. Ramharter, E. Joekes, and S. Bélard. Please ensure that the full contributions of each author are acknowledged in the "Add/Edit/Remove Authors" section of our submission form.

4) We note that there is identifying data in the Supporting Information file(s) Database.xlsx. Prior to sharing human research participant data, authors should consult with an ethics committee to ensure data are shared in accordance with participant consent and all applicable local laws.

 - Name, initials, physical address

 - Ages more specific than whole numbers

 - Internet protocol (IP) address

 - Specific dates (birth dates, death dates, examination dates, etc.)

 - Contact information such as phone number or email address

 - Location data

 - ID numbers that seem specific (long numbers, include initials, titled “Hospital ID”) rather than random (small numbers in numerical order)

Additional guidance on preparing raw data for publication can be found in our Data Policy: https://journals.plos.org/plosntds/s/data-availability#loc-human-research-participant-data-and-other-sensitive-data

You may also find guidance in the following article: http://www.bmj.com/content/340/bmj.c181.long.

Please remove or anonymize all personal information: Specific dates, and ID numbers that seem specific, ensure that the data shared are in accordance with participant consent, and re-upload a fully anonymized data set. Please note that spreadsheet columns with personal information must be removed and not hidden as all hidden columns will appear in the published file.

**Reviewers' comments:** Reviewer's Responses to Questions

**Key Review Criteria Required for Acceptance?**

**Methods**

-Are the objectives of the study clearly articulated with a clear testable hypothesis stated?

-Is the study design appropriate to address the stated objectives?

-Is the population clearly described and appropriate for the hypothesis being tested?

-Is the sample size sufficient to ensure adequate power to address the hypothesis being tested?

-Were correct statistical analysis used to support conclusions?

-Are there concerns about ethical or regulatory requirements being met?

Reviewer #1: All data are clearly presented

Reviewer #2: Are the objectives of the study clearly articulated with a clear testable hypothesis stated?

Yes.

-Is the study design appropriate to address the stated objectives?

Yes.

-Is the population clearly described and appropriate for the hypothesis being tested?

Yes.

-Is the sample size sufficient to ensure adequate power to address the hypothesis being tested?

Yes.

-Were correct statistical analysis used to support conclusions?

Yes.

-Are there concerns about ethical or regulatory requirements being met?

Yes.

Reviewer #3: The paper present a simple objective and followed good procedure. However, the procedure need expansion

**Results**

-Does the analysis presented match the analysis plan?

-Are the results clearly and completely presented?

-Are the figures (Tables, Images) of sufficient quality for clarity?

Reviewer #1: Yes

Reviewer #2: Does the analysis presented match the analysis plan?

Yes.

-Are the results clearly and completely presented?

Yes.

-Are the figures (Tables, Images) of sufficient quality for clarity?

Yes.

Reviewer #3: The result is clearly stated

**Conclusions**

-Are the conclusions supported by the data presented?

-Are the limitations of analysis clearly described?

-Do the authors discuss how these data can be helpful to advance our understanding of the topic under study?

-Is public health relevance addressed?

Reviewer #1: Yes, definitely. Very important conclusions for improvement of control of schistosmiasis

Reviewer #2: -Are the conclusions supported by the data presented?

Yes.

-Are the limitations of analysis clearly described?

Yes.

-Do the authors discuss how these data can be helpful to advance our understanding of the topic under study?

Yes.

-Is public health relevance addressed?

Yes.

Reviewer #3: Fair

**Editorial and Data Presentation Modifications?**

Reviewer #1: None

Reviewer #2: Accept

Reviewer #3: sentences in line 213 is not clear to me.

**Summary and General Comments**

Reviewer #1: Well written article with important epidemiologic and clinical significance

Reviewer #2: This study aimed to detect urinary tract pathology in symptomatic UGS patients in an endemic area and to correlate the findings with demographic, clinical, and parasitological factors. The authors reported the urinary tract pathology related to UGS in Gabon for the first time. This study has a chance to primarily guide local schistosomiasis control and can add valuable information to control strategies and individual patient care on a global level. Scientific sound of text and references are adequate.

Reviewer #3: The paper present a simple method in the work using known tool to test for S. hematobium among children. However, there was no any novelty in the work beyond looking at underage children. There should be additional study using more tools e.g. molecular diagnostics, RDT to further validate the claim of 'potentially irreversible pathology' among children. for an infection to be irreversible, it mean that genetic alteration had occurred and no evidence that such had taken place. Also, the result is too little for publication except case report

PLOS authors have the option to publish the peer review history of their article (what does this mean? ). If published, this will include your full peer review and any attached files.

**Do you want your identity to be public for this peer review?** For information about this choice, including consent withdrawal, please see our Privacy Policy .

Reviewer #1: No

Reviewer #2: No

Reviewer #3: No

---

## [Editor Report · Decision Letter 1]

14 Mar 2025

Dear Dr Remppis,

We are pleased to inform you that your manuscript 'Point-of-care ultrasound reveals extensive pathology in Gabonese preschool-age children with urogenital schistosomiasis' has been provisionally accepted for publication in PLOS Neglected Tropical Diseases.

Best regards,

Jong-Yil Chai

Section Editor

Shaden Kamhawi

co-Editor-in-Chief

Paul Brindley

co-Editor-in-Chief

The revised manuscript has properly addressed the comments given by the reviewers' comments and is now acceptable for publication.

---

## [Editor Report · Acceptance letter]

Dear Dr Remppis,

We are delighted to inform you that your manuscript, "Point-of-care ultrasound reveals extensive pathology in Gabonese preschool-age children with urogenital schistosomiasis," has been formally accepted for publication in PLOS Neglected Tropical Diseases.

Best regards,

Shaden Kamhawi

co-Editor-in-Chief

Paul Brindley

co-Editor-in-Chief
